# Epidemiological Situation of Glanders in the State of Pará, Brazil

**DOI:** 10.3390/pathogens12020218

**Published:** 2023-01-31

**Authors:** Ana Paula Vilhena Beckman Pinho, Fernando Ferreira, Jeferson Jacó Fuck, Jefferson Pinto de Oliveira, Ricardo Augusto Dias, José Henrique Hildebrand Grisi-Filho, Marcos Bryan Heinemann, Evelise Oliveira Telles, José Soares Ferreira Neto

**Affiliations:** 1Agency of Sanitary Defence of Agriculture and Livestock of the state of Pará, Travessa Mariz de Barros, 1184, Belém CEP 66080-008, PA, Brazil; 2Faculty of Veterinary Medicine and Animal Science, University of São Paulo, Avenida Professor Doutor Orlando Marques de Paiva, 87, São Paulo CEP 05508-270, SP, Brazil; 3Ministry of Agriculture, Livestock and Food Supply, Esplanada dos Ministérios, Bloco D, Brasília CEP 70043-900, DF, Brazil

**Keywords:** glanders, equids, prevalence, risk factor, Pará, Brazil

## Abstract

Glanders is an anthropozoonosis caused by the bacteria *Burkholderia mallei*, affecting mainly equids. It has been eradicated in North America, Australia, and Western Europe, but continues to occur sporadically in countries in Asia, Africa, the Middle East, and South America. Its notification is mandatory by the World Organization for Animal Health. After 30 years, the disease reappeared in Brazil in 1999 and, thereafter, 1,413 outbreaks have been reported. However, the epidemiological situation of the disease in the country is not adequately known. Thus, 2718 animals from 654 properties in the state of Pará were randomly selected by sampling and examined using a serial protocol with Complement Fixation and Western Blot serological tests. The prevalence of properties infected with glanders in the state was estimated at 1.68% [0.84; 3.33] and of seropositive animals at 0.50% [0.27; 0.94]. The introduction of animals was individualized as a risk factor for disease introduction in the properties (OR = 5.9 [1.4; 25.5]). Despite the low prevalence of infected properties and seropositive animals, the state must review actions to fight the disease, considering that the strategies implemented have not affected the endemic balance of the disease. This process must involve all public and private agents interested in the topic.

## 1. Introduction

Glanders is a disease known since ancient Greek and Roman times that mainly affects equids. Its etiologic agent is the bacteria *Burkholderia mallei*, which causes nodules and ulcerations in the respiratory tract and lungs of affected animals. It is a rare disease in humans and can affect veterinarians, people who work with horses, and laboratory personnel [1]. The disease spread throughout the world mainly due to the movement of equids during war periods. Transmission occurs by ingestion of contaminated water or food, contaminated fomites, and infectious aerosols [1].

It has been eradicated in North America, Australia, and Western Europe, but continues to occur sporadically in countries in Asia, Africa, the Middle East, and South America. According to the World Organization for Animal Health (WOAH), it is a notifiable disease [1].

In Brazil, glanders was described for the first time in 1811, probably as a result of the importation of infected animals from Europe [2,3]. Since then, it was reported in animals and humans in several Brazilian regions until the early 1960s, when it seemed to have disappeared from the country until it was detected again in 1999 in the states of Pernambuco and Alagoas [4].

Since 1999, Brazil notified 1413 glanders outbreaks to the WOAH [5], but good quality data on the epidemiological situation of the disease in the Brazilian states are scarce. Only in the Distrito Federal, the smallest Brazilian Federative Unit, located in the center of the country, a well-planned prevalence study was conducted in 2010, covering its entire territory with a sample targeted at traction equids [6]. The author did not detect any positive animals, but using the beta distribution, he calculated the upper limit of the confidence interval for the prevalence of infected properties to be 0.85%

Other Brazilian studies reported simple proportions of “positive animals in routine tests for movement/tested animals” [7,8] or just the “number of positive animals in routine tests for movement or of outbreaks in a given period” [9,10,11].

The state of Pará has a size of 1.2 million km^2^ and about 9 million inhabitants. Although the main economic activity is mining, beef production is of great importance and there are about 24 million cattle in the state.

According to the Agency of Sanitary Defence of Agriculture and Livestock of the State of Pará (Agência de Defesa Agropecuária do Estado do Pará—ADEPARÁ), there are currently around 550,000 equids in the state, distributed across approximately 89,000 properties. Since 2005, ADEPARÁ has detected 53 glanders outbreaks (Figure 1), but there are no good quality data on the epidemiological situation of the disease in the state which would allow an adequate case definition and improved actions for fighting glanders in Pará [12,13].

Thus, the objective of the present study was to estimate the prevalence of glanders-infected properties and infected animals in the state of Pará. In addition, it is also intended to individualize the risk factors associated with the disease.

## 2. Materials and Methods

### 2.1. Study Design

The study was conducted by ADEPARÁ with the support of the Ministry of Agriculture, Livestock and Food Supply (Ministério da Agricultura, Pecuária e Abastecimento—MAPA) and the Collaborating Center for Animal Health of the Faculty of Veterinary Medicine and Animal Science of the University of São Paulo (FMVZ-USP). Fieldwork was concluded in June 2019 by ADEPARÁ.

To capture eventual internal heterogeneities, the state was divided into regions according to the predominance of the typology of rural properties, equine marketing practices, management practices, and exploration purposes, respecting ADEPARÁ’s operational capacity.

In each region, a sample of farms and animals was randomly selected. The animals were submitted to serological diagnosis of glanders by Complement Fixation (CF) and Western Blot (WB) serial testing, according to Ordinance no. 35 of the MAPA [13]. The tests were performed at the National Reference Laboratory, located in Recife, state of Pernambuco (PE).

A questionnaire was administered in the selected properties to collect data on property characteristics (location, equid herd, breeding type and purpose, presence of wetlands), animal management (reproduction system, animal introduction, participation in livestock events), and sanitary practices (testing for glanders, sharing fomites and needles, and veterinary care), to verify possible associations with the disease.

The results of the questionnaires and serology analyses were entered into a database and analyzed at the Laboratory of Epidemiology and Biostatistics of the FMVZ-USP Collaborating Center for Animal Health.

### 2.2. Sampling

For each region, a two-stage sample was used. In the first stage, an established number of properties with equids aged six months or over was randomly selected based on the ADEPARÁ register. Those who could not participate in the sample were replaced through a new draw. In each selected farm, a minimum number of equids aged six months or over were examined to classify the farm as infected or non-infected with glanders (second sampling stage). The animals were randomly selected and a 10 mL blood sample was collected from each of them for serological diagnosis.

In view of the great operational difficulties foreseen for conducting the field work in the northern region of the state, the sample for the first stage, calculated for simple random samples [14], had to guarantee flexibility through the establishment of minimum and maximum number of farms to be sampled in each region.

The assumptions for calculating the minimum sample value were: estimated prevalence = 0.20, precision = 0.08, confidence level = 0.95, and population size = 88,602 properties, resulting in a sample of 97 properties.

The assumptions for calculating the maximum sample value were: estimated prevalence = 0.20, precision = 0.05, confidence level = 0.95, and population size = 88,602 properties, resulting in a sample of 246 properties.

Calculations were performed using the Epitools software [15].

The sensitivity and specificity of the diagnostic protocol used were 94.86% and 99.98%, respectively. These values were calculated considering the use of serial CF and WB tests. Calculations were performed using the Epitools software [16], using sensitivity and specificity values of 98.00% and 96.40% for CF and 96.80% and 99.40% for WB [17].

Thus, aggregate sensitivity and specificity were calculated by sampling stage using 94.86% sensitivity, 99.98% specificity, and 15% intra-herd prevalence.

The number of animals examined in each farm ensured minimum aggregate sensitivity and specificities of 89.4% and 99.7%, respectively. The operational animal sampling protocol used in the properties is shown in Table 1. Calculations were performed using the Epitools software [18].

### 2.3. Data Processing

Apparent prevalence of infected properties and seropositive animals and their respective confidence intervals were calculated for each region and state according to Dean et al. [19]. The prevalence of infected properties and seropositive animals in the state and animal prevalence in the regions were weighted according to Dohoo et al. [20]. The weight of each property to calculate the prevalence of infected properties in the state was given by:$$P1=\frac{\mathrm{Properties}\;\mathrm{in}\;\mathrm{the}\;\mathrm{region}}{\mathrm{Properties}\;\mathrm{sampled}\;\mathrm{in}\;\mathrm{the}\;\mathrm{region}}$$

The weight of each animal to calculate the prevalence of seropositive animals in the state was given by: $$P2=\frac{\mathrm{Equids}\;\geq \;6\;\mathrm{months}\;\mathrm{in}\;\mathrm{the}\;\mathrm{property}}{\mathrm{Equids}\;\geq \;6\;\mathrm{months}\;\mathrm{sampled}\;\mathrm{in}\;\mathrm{the}\;\mathrm{property}}\times \frac{\mathrm{Equids}\;\geq \;6\;\mathrm{months}\;\mathrm{old}\;\mathrm{in}\;\mathrm{the}\;\mathrm{region}}{\mathrm{Equids}\;\geq \;6\;\mathrm{months}\;\mathrm{sampled}\;\mathrm{in}\;\mathrm{the}\;\mathrm{region}}$$

In the expression above, the first term refers to the weight of each animal to calculate the prevalence of seropositive animals in the regions.

Considering results from the entire state, two groups of properties were formed—infected and non-infected with glanders—which, when compared with one another regarding the variables surveyed in the questionnaires, allowed measuring the strength of the association between these variables and the presence of the disease. A first exploratory analysis of the data (univariate) was conducted to select those with *p* ≤ 0.20 for the χ^2^ test and subsequent multivariate logistic regression [21].

The final multivariable model was built using the stepwise forward method, with sequential inclusion of the most significant variables in the univariate analysis. A variable was kept in the model when it improved the fit measured by the maximum likelihood ratio test. At the same time, the variable coefficient needed to be statistically different from zero (*p* < 0.05, Wald test). The goodness of fit of the final model was assessed by the ROC curve [20]. All calculations were performed in the R CORE TEAM software [22].

## 3. Results

The state was divided into four regions. Table 2 shows registration and sample data. Figure 2 shows the spatial location and sanitary status for Glanders of the sampled properties. The prevalence of infected properties is shown in Table 3. The results for the prevalence of seropositive animals are shown in Table 4 and the final model for risk factors associated with glanders in the state of Pará in Table 5.

## 4. Discussion

Table 3 and Table 4 and Figure 2 show the very low prevalence of infected properties and seropositive animals for glanders in the state of Pará. Although the data in Table 2 suggest a greater proportion of infected properties in Region 2, a statistically significant difference was found only when comparing the prevalence in regions 2 and 3 (*p* = 0.03 for Fisher’s exact test).

Considering the most likely value and the lower limit of the estimated prevalence of seropositive animals for the state (0.5% and 0.27%, Table 4) and the previously mentioned sensitivity and specificity of the diagnostic procedure (94.86% and 99.98%, respectively), the most likely positive and negative predictive values of serial CF and WB tests were calculated at 96.02% and 99.97%, respectively, with worst-case scenarios of 92.61% and 99.99%. Thus, considering the epidemiological situation of glanders in Pará, there is high confidence in the protocol adopted for serological diagnosis, as it classifies infected and healthy animals with a high probability of success.

The calculation of positive and negative predictive values depends on the estimated prevalence and, therefore, the Brazilian states should conduct epidemiological studies to clarify the situation of glanders in their territories for rational management by the National Equid Health Program (Programa Nacional de Sanidade dos Equídeos—PNSE). However, it is fundamental that these studies be standardized, i.e., carried out with the same methodology.

Inferring the estimated prevalence of infected properties (Table 3) to the existing number of properties with equids in Pará (Table 2), the state had 1489 [744; 2950] properties infected with glanders in 2019. Considering that the annual mean number of outbreaks detected in the state between 2005–2019 was 1.73 (Figure 1), the mean sensitivity of the surveillance system for glanders in the period ranged from 0.06–0.23%, with 0.10% as the most likely number, a very low value and probably insufficient for changing the endemic balance of the disease, which indicates that the PNSE actions are not reaching the expected objectives, requiring a reassessment that involves all public and private agents interested in the subject in the process [23,24].

The key elements in this discussion are: (1) appreciation of the results presented here; (2) clear definition of the objective of the program: control or eradication; (3) recognition that glanders is a zoonosis; (4) acknowledgment of the existence of successful international experience for equid health programs, with shared responsibilities and costs between the production chain and the Official Veterinary Service [25]; and (5) the need for mechanisms to assess the effectiveness of implemented actions.

The final logistic regression model revealed that properties that introduced animals had a greater chance of being infected with glanders (Table 5). It is important to highlight the small number of infected properties, resulting in a very wide confidence interval for the odds ratio estimate (Table 5). However, the introduction of animals, naturally without prior testing for glanders, has high plausibility as a risk factor for the disease, considering the mechanisms of glanders transmission.

## 5. Conclusions

Equid breeders in the state of Pará must be informed that glanders is a rare disease in the state and that its spread is associated with the introduction of animals into herds without prior testing. The state must review its actions to combat the disease, considering that the strategies implemented so far have not affected the endemic balance of the disease. This process must involve all public and private agents interested in the topic.

## Figures and Tables

**Figure 1 pathogens-12-00218-f001:**
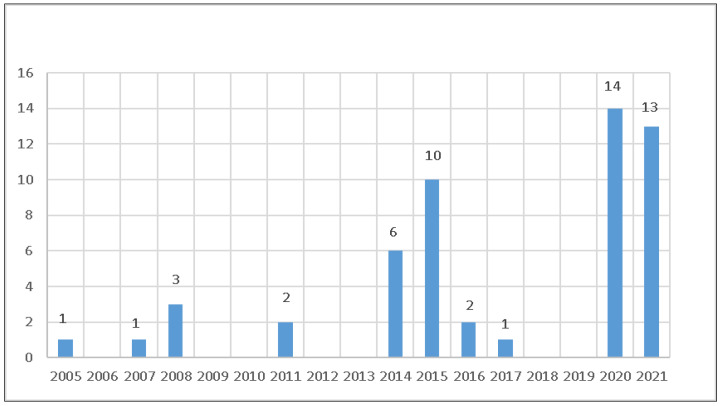
Number of glanders outbreaks detected in the state by the Agency of Sanitary Defence of Agriculture and Livestock of the state of Pará, Brazil, in 2005–2021. Source: https://indicadores.agricultura.gov.br/saudeanimal/index.htm (accessed on 4 December 2022).

**Figure 2 pathogens-12-00218-f002:**
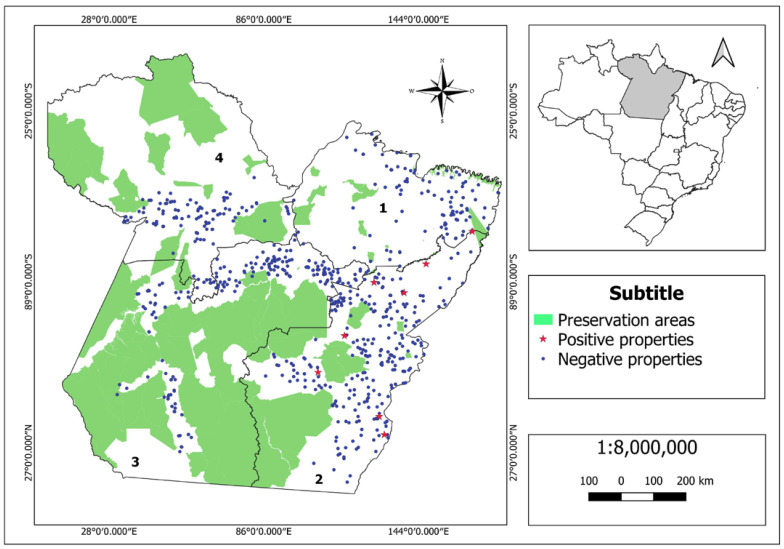
Map of the state of Pará showing the preservation areas (environmental protection areas, biological and extractive reserves, and indigenous lands), and information about the study of glanders: the division of the state into 4 regions and the geographic location and the sanitary condition of the properties sampled/tested (left map). The map on the right shows the location of the state of Pará in Brazil.

**Table 1 pathogens-12-00218-t001:** Sampling planning for animals randomly selected in each farm.

Number of Equidae Aged ≥6 Months on the Farm	Number of Equidae Aged ≥6 Months Sampled on the Farm
1–10	all
11–15	10
16–20	11
21–30	12
31–60	13
61–180	14
≥181	15

**Table 2 pathogens-12-00218-t002:** Registration and sample data of the study on glanders in the equid population of the state of Pará, Brazil.

Region	Number of Properties with Equidae	Number of Equidae Aged ≥6 Months	Number of Properties with Equidae Sampled	Number of Equidae Aged ≥6 Months Sampled
1	11,132	88,981	102	486
2	48,426	297,312	260	1080
3	17,481	91,227	178	733
4	11,563	59,143	114	419
**Pará**	**88,602**	**536,663**	**654**	**2718**

**Table 3 pathogens-12-00218-t003:** Prevalence of properties infected with glanders in the state of Pará, Brazil.

Region	Properties		Infected Farm Prevalence (%)	Confidence Interval 95% (%)	
	Positive	Sampled		Lower Limit	Upper Limit
1	0	102	0.00	0.00	2.87 *
2	8	260	3.08	1.57	5.93
3	0	178	0.00	0.00	1.65 *
4	0	114	0.00	0.00	2.57 *
**Pará**	**8**	**654**	**1.68**	**0.84**	**3.33**

* calculated by the ꞵ distribution and Monte Carlo simulation.

**Table 4 pathogens-12-00218-t004:** Prevalence of glanders-seropositive animals in the state of Pará, Brazil.

Region	Equidae ≥ 6 Months		Prevalence of Seropositive Animals (%)	Confidence Interval 95% (%)	
	Seropositive	Sampled		Lower Limit	Upper Limit
1	0	486			
2	10	1080	0.92	0.43	1.71
3	0	733			
4	0	419			
**Pará**	**10**	**2718**	**0.50**	**0.27**	**0.94**

**Table 5 pathogens-12-00218-t005:** Final logistic regression model for glanders-associated risk factors in the state of Pará, Brazil.

Variable	OR	Confidence Interval 95% (%)		*p* Value
		Lower Limit	Upper Limit	
Introducing Equidae to the Farm				
no (basic category)				
yes	5.94	1.39	25.47	0.016

## Data Availability

The original databases can be requested from the corresponding author (JSFN).

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
