# Peer review of "Epidemiological Situation of Glanders in the State of Pará, Brazil"

_pathogens, 2023, doi:10.3390/pathogens12020218_

Round 1

Reviewer 1 Report

The information that is provided in the manuscript is important since information regarding glanders is limited.

Minor remarks

Were there animals other than horses? If no, I suggest to change to horses instead of equids or animals.

Line 34 – please correct “since ancient the time of the Greeks and Romans”

Line 36 – please elaborate and add reference “It can be transmitted to humans.”

I suggest adding just one sentence in the introduction about Para, to make it clear to readers, who are not familiar, although a map is included later in the manuscript.

Line 61 – please explain the difference between 47 and the number the figure 1 (53).

Line 94 – please delete the dot and space before the sub-title.

Table 1. please change Todos to all.

Maybe you could add these new references on glanders in Brazil:

Current status of glanders in Brazil: recent advances and challenges

Rinaldo Aparecido Mota 1José Wilton Pinheiro Junior 2

Glanders and brucellosis in equids from the Amazon region, Brazil.

Resende CF, Santos AMD, Filho PMS, de Souza PG, Issa MA, Filho MBC, Victor RM, Câmara RJF, Gonçalves GP, Lima JG, Maciel E Silva AG, Leite RC, Reis JKPD.

Author Response

Response to Reviewer 1 Comments

Note: in the attached revised paper, the corrections are in red to facilitate editing

Point 1: Were there animals other than horses? If no, I suggest to change to horses instead of equids or animals.

Response 1: We sampled and tested horses, mules and donkeys.

Point 2: line 34 – please correct “since ancient the time of the Greeks and Romans”

Response 2: We corrected to "since ancient time of the Greeks and Romans".

Point 3 Line 36 – please elaborate and add reference “It can be transmitted to humans.”

Response 3: We have enriched the text to "It is a rare disease in humans and can affect veterinarians, people who work with horses, and laboratory personnel [1]” .

Point 4: I suggest adding just one sentence in the introduction about Para, to make it clear to readers, who are not familiar, although a map is included later in the manuscript.

Response 4:  We have included the following paragraph:

The state of Pará has 1.2 million km2 and about 9 million inhabitants. Although the main economic activity is mining, beef production is of great importance for the state, which has about 24 million cattle.

Point 5: Line 61 – please explain the difference between 47 and the number the figure 1 (53).

Response 5: Very sorry for the error in the summation. We corrected it to 53

Point 6: Line 94 – please delete the dot and space before the sub-title.

Response 6: We delete the space.

Point 7: Table 1. please change Todos to all.

Response 7: Done.

Point 8:  Maybe you could add these new references on glanders in Brazil: Mota, R.A., Junior, J.W.P.  Current status of glanders in Brazil: recent advances and challenges.  Braz. J. Microbiol., 2022, v.53, p.2273–2285.

Response 8: Done. Thanks for the suggestion. See line 284.

IMPORTANT: We have corrected an error in the estimation of the Surveillance System Sensitivity in lines 94 and 95 of the revised paper.

Reviewer 2 Report

A seroprevalence study detecting antibodies against Burkholderia mallei (glanders) has been conducted in a northern province of Brazil. B. mallei is an important pathogen for the equine industry as many countries are negative for this pathogen, however, fear a re-introduction with transportation of horses across borders. The study is well composed and appropriate tests have been used. 

The scientific English is quite good, however, some formulations need a bit of smoothing in certain areas, which will make it easier to understand for the reader.  

in greater detail: a literature reference (6) refers to a thesis in portugues and is not accessible to me. Apparently it describes an earlier study on glanders prevalence, and I would encourage the authors to expand a little more, (line 52) 

line 52: draft animals - ?

line 78: I don't understand '.. regions by type of production and commercialization'

line 81: should it be 'sample set'?

line 83: what is MAPA

line 99: sampling strategy- what were the minimum numbers of horses sampled, a percentage maybe?

line 104: are you referring to sample size (min-max)?

line 149: how does Fig 2show 'sanitary condition' of sample

Figure 2 needs letters in each panel and an arrow that shows the highlighted and magnified Para area in the map of Brazil. 

L163: I did not understand 'Introduction of Animals' in Table 5

In general: have you considered specificity of the test - cross-reactivity? 

Author Response

Response to Reviewer 2 Comments

Note: in the attached revised paper, the corrections are in red to facilitate editing

Point 1: The scientific English is quite good, however, some formulations need a bit of smoothing in certain areas, which will make it easier to understand for the reader.

Response 1: We have revised the text, trying to make it easier to understand.

Point 2: in greater detail: a literature reference (6) refers to a thesis in portugues and is not accessible to me. Apparently it describes an earlier study on glanders prevalence, and I would encourage the authors to expand a little more, (line 52)

Response 2:  The master's thesis is available at https://repositorio.unb.br/bitstream/10482/10231/1/2011_DaniellaDianeseAlvesdeMoraes.pdf. We have improved the text to: “Only in the Distrito Federal, the smallest Brazilian Federative Unit, located in the center of the country, a well-planned prevalence study was conducted in 2010, covering its entire territory with a sample targeted at traction equids [6]. The author did not detect any positive animals, but using the beta distribution, he calculated the upper limit of the confidence interval for the prevalence of infected properties to be 0.85%”.

Point 3 line 52: draft animals - ?

Response 3:  We corrected it to: “traction equids”

Point 4: line 78: I don't understand '.. regions by type of production and commercialization'

Response 4:  We have improved the text to : " We improved the text to : "the state was divided into regions according to the predominance of the typology of rural properties, equine marketing practices….".

Point 5: line 81: should it be 'sample set'?

Response 5: We improved the text to: "In each region, a sample of farms and animals was randomly selected. The animals were submitted to serological diagnosis...., state of Pernambuco."

Point 6: line 83: what is MAPA

Response 6: It is the acronym of Ministry of Agriculture, Livestock and Food Supply (Ministério da Agricultura, Pecuária e Abastecimento - MAPA), described on line 78-80 of the revised paper.

Point 7: line 99: sampling strategy- what were the minimum numbers of horses sampled, a percentage maybe?

Response 7: It is not a percentage, but the minimum number of animals that must be examined to classify the farm as infected or not.

Point 8: line 104: are you referring to sample size (min-max)?

Response 8: Yes. Trying to improve the text, we changed it to: “... had to guarantee flexibility through the establishment of minimum and maximum number of farms to be sampled in each region”.

Point 9: line 149: how does Fig 2show 'sanitary condition' of sample

Response 9: We change the text to: “Figure 2 shows the spatial location and sanitary status for Glanders of the sampled properties.”

Point 10: Figure 2 needs letters in each panel and an arrow that shows the highlighted and magnified Para area in the map of Brazil.

Response 10: We inform in the figure's title where each map is.

Point 11: L163: I did not understand 'Introduction of Animals' in Table 5

Response 11: Introduction of animals involves the purchase, loan, donation, or any other practice that results in the introduction of equids to the property. We have tried to improve the variable description in the table to: “introducing equidae to the farm”.

Point 12: In general: have you considered specificity of the test - cross-reactivity? 

Response 12: The individual sensitivity and specificity of each test (CF and WB) were considered to: 1) calculate the sensitivity and the specificity of the serial testing protocol, 2) calculate the true prevalence of seropositive animals, needed to estimate the positive and negative predictive values. The sensitivity and specificity of the serial testing protocol were used to estimate the sample size of animals to be tested within the farms.

IMPORTANT: We have corrected an error in the estimation of the Surveillance System Sensitivity in lines 94 and 95 of the revised paper.